# mRNA-1273 bivalent (original and Omicron) COVID-19 vaccine effectiveness against COVID-19 outcomes in the United States

Hung Fu Tseng [1,2] ✉, Bradley K. Ackerson [1], Lina S. Sy [1], Julia E. Tubert[1], Yi Luo [1], Sijia Qiu[1], Gina S. Lee [1], Katia J. Bruxvoort [1,3], Jennifer H. Ku[1], Ana Florea [1], Harpreet S. Takhar[1], Radha Bathala[1], Cindy Ke Zhou[4], Daina B. Esposito[4], Morgan A. Marks[4], Evan J. Anderson [4], Carla A. Talarico[4,5] & Lei Qian[1]

The bivalent (original and Omicron BA.4/BA.5) mRNA-1273 COVID-19 vaccine was authorized to offer broader protection against COVID-19. We conducted a matched cohort study to evaluate the effectiveness of the bivalent vaccine in preventing hospitalization for COVID-19 (primary outcome) and medically attended SARS-CoV-2 infection and hospital death (secondary outcomes). Compared to individuals who did not receive bivalent mRNA vaccination but received ≥2 doses of any monovalent mRNA vaccine, the relative vaccine effectiveness (rVE) against hospitalization for COVID-19 was 70.3% (95% confidence interval, 64.0%–75.4%). rVE was consistent across subgroups and not modified by time since last monovalent dose or number of monovalent doses received. Protection was durable ≥3 months after the bivalent booster. rVE against SARS-CoV-2 infection requiring emergency department/urgent care and against COVID-19 hospital death was 55.0% (50.8%–58.8%) and 82.7% (63.7%–91.7%), respectively. The mRNA-1273 bivalent booster provides additional protection against hospitalization for COVID-19, medically attended SARS-CoV-2 infection, and COVID-19 hospital death.

As of May 2023, COVID-19 has resulted in >6.1 million hospitalizations and >1.1 million deaths in the United States[1]. Monovalent vaccines designed to protect against the original strain of SARS-CoV-2 were highly effective in reducing COVID-19 morbidity and mortality[2,3]. However, effectiveness of vaccines against SARS-CoV-2 infection has declined over time due to waning immunity as well as the emergence of SARS-CoV-2 viral variants leading to immune evasion from naturally acquired and/or vaccine-elicited immunity[4–6]. To address this concern, mRNA BA.4/BA.5 bivalent booster vaccines, containing equal amounts of spike protein sequences for Omicron subvariants BA.4/BA.5 and original SARS-CoV-2 spike proteins, were developed.

Pre-clinical data suggests that BA.4/BA.5-containing mRNA vaccine provides greater neutralizing activity against BA.4/BA.5 and other emerging SARS-CoV-2 Omicron sublineages compared to original mRNA vaccines[7,8]. Based on these factors and in anticipation of an increase in cases during the winter respiratory season[9], the U.S. Food and Drug Administration (FDA) authorized the Moderna and Pfizer-BioNTech bivalent mRNA vaccines on 31 August 2022 for individuals ≥18 years of age who had received their last of at least two monovalent doses at least 2 months earlier[10,11]. Subsequently, as of 18 April 2023, the Moderna and Pfizer-BioNTech bivalent mRNA vaccines were authorized for all doses administered to individuals aged 6 months of age and older[12].

[1]Department of Research and Evaluation, Kaiser Permanente Southern California, Pasadena, CA 91101, USA. [2]Department of Health Systems Science, Kaiser Permanente Bernard J. Tyson School of Medicine, Pasadena, CA 91101, USA. [3]Department of Epidemiology, University of Alabama at Birmingham, Birmingham, AL 35233, USA. [4]Moderna Inc., Cambridge, MA 02139, USA. [5]AstraZeneca, Gaithersburg, MD 20878, USA. ✉e-mail: Hung-Fu.X.Tseng@kp.org

Recent real-world studies in the United States have demonstrated improved effectiveness of BA.4/BA.5-containing mRNA bivalent boosters against COVID-19 outcomes such as hospitalization, symptomatic infection, and death relative to monovalent vaccination and no vaccination[10,13–15]. However, these early studies evaluated VE against non-medically attended symptomatic infection[10,15], medically attended infection[13], or hospitalization during September to early December 2022 when BA.5 and BQ.1/BQ.1.1 were predominant[5,13,16].

We conducted a matched prospective cohort study at Kaiser Permanente Southern California (KPSC) with follow-up through 01/31/2023 to assess the effectiveness of the bivalent (original and Omicron BA.4/BA.5) mRNA-1273 COVID-19 vaccine in preventing hospitalization for COVID-19 (primary outcome), medically attended SARS-CoV-2 infection, and COVID-19 hospital death in a real-world setting during a period of BA.5 and BQ.1 predominance and subsequent emergence of XBB sublineages.

## Results

The study included 290,292 recipients of the bivalent mRNA-1273 vaccine (follow-up time for primary outcome analysis: median 2.46 months, maximum 4.57 months) and 580,584 age-, sex-, and race/ethnicity-matched individuals who received ≥2 doses of monovalent mRNA COVID-19 vaccine only (follow-up time for primary outcome analysis: median 1.45 months, maximum 4.57 months) (Fig. 1). Detailed distribution of sociodemographic variables and baseline characteristics of the two cohorts is presented in Table 1. The bivalent vaccine group and the ≥2 monovalent mRNA vaccine group had different distributions (ASD > 0.1) for number of outpatient and virtual visits, preventive care, number of monovalent vaccines prior to index date, time between latest monovalent vaccine and index date, and medical center area. The study also included 204,655 individuals who never received any COVID-19 vaccine (follow-up time for primary outcome analysis: median 2.33 months, maximum 4.57 months) that were matched up to 1:1 to the bivalent cohort on age-, sex-, and race/ethnicity (Fig. 1). The detailed distribution of sociodemographic variables and baseline characteristics of the two cohorts is presented in Supplementary Table 1. They had different distributions (ASD > 0.1) of age and race/ethnicity due to inability to completely match bivalent boosted with unvaccinated persons, as described in the Methods section. As expected, the bivalent vaccine group and the COVID-19 unvaccinated group differed across several variables, including body mass index, smoking, Charlson comorbidity score, frailty index, kidney disease, lung disease, diabetes, immunocompromised (IC) status, history of SARS-CoV-2 infection, history of SARS-CoV-2 molecular test, number of outpatient and virtual visits, preventive care, Medicaid status, neighborhood median household income, and medical center area.

Figure 2a shows the adjusted relative vaccine effectiveness (rVE) of the bivalent vaccine, compared to the ≥2 monovalent mRNA vaccine group, against hospitalization for COVID-19 disease (defined in the outcomes of interest section under Methods); case numbers and incidence rates are presented in Supplementary Table 2. Overall, the rVE against hospitalization for COVID-19 was 70.3% (95% confidence interval [CI]: 64.0%–75.4%). The rVE was slightly lower in the 45–64-year-old group (56.2%, 95% CI: 22.5%–75.2%), but in general was consistent across age, sex, and race/ethnicity group. The rVE was 64.7% (95% CI: 44.0%–77.7%) in IC individuals, compared to 71.3% (95% CI: 64.5%–76.7%) in immunocompetent individuals. The rVE was 60.7% (95% CI: 33.8%–76.7%) in individuals with a history of SARS-CoV-2 infection and 71.1% (95% CI: 64.6%–76.5%) in those without known history. There was no apparent waning in protection against hospitalization for COVID-19, with rVE remaining at 79.6% (95% CI: 43.2%–92.7%) at ≥3 months since bivalent vaccination, although the confidence interval was relatively wide. The number of prior monovalent vaccine doses received and the time since the last dose of monovalent vaccine did not appear to substantially modify rVE.

Figure 2b shows the adjusted absolute vaccine effectiveness (VE) of the bivalent vaccine, compared to the COVID-19 unvaccinated group, against hospitalization for COVID-19 disease; case numbers and incidence rates are presented in Supplementary Table 3. Overall, the VE against hospitalization for COVID-19 was 82.8% (95% CI: 78.8%–86.0%). It was generally consistent across age, sex, and race/ethnicity group. The VE was 71.8% (95% CI: 48.8%–84.5%) in IC individuals, compared to 84.1% (95% CI: 80.1%–87.4%) in immunocompetent individuals. The VE was 68.3% (95% CI: 45.4%–81.6%) in individuals with history of SARS-CoV-2 infection and 84.3% (95% CI: 80.3%–87.5%) in those without known history. There was no apparent waning in protection against hospitalization for COVID-19 with VE remaining at 75.5% (95% CI: 43.8%–89.3%) at ≥3 months since bivalent vaccination.

Cumulative incidence of hospitalization for COVID-19 was significantly higher in the two comparator groups compared to the bivalent vaccine group (log-rank test p < 0.0001; Figs. 3a and 3b). At the end of follow-up, the cumulative incidence was 0.09%, 0.22%, and 0.27% in the bivalent vaccine, ≥2 monovalent mRNA vaccine, and COVID-19 unvaccinated groups, respectively.

Figure 4a shows the adjusted rVE of the bivalent vaccine, compared to the ≥2 monovalent mRNA vaccine group, in preventing medically attended SARS-CoV-2 infection and COVID-19 hospital death as the secondary outcomes; case numbers and incidence rates are presented in Supplementary Table 4. The rVE against medically attended SARS-CoV-2 infection in all care settings and in emergency department/urgent care (ED/UC) settings was 35.9% (95% CI: 32.7%–39.0%) and 55.0% (95% CI: 50.8%–58.8%), respectively. The rVE against COVID-19 hospital death was 82.7% (95% CI: 63.7%–91.7%).

Figure 4b shows the adjusted absolute VE of the bivalent vaccine, compared to the COVID-19 unvaccinated group, in preventing medically attended SARS-CoV-2 infection and COVID-19 hospital death; case numbers and incidence rates are presented in Supplementary Table 5. The VE against medically attended SARS-CoV-2 infection in all care settings and in ED/UC settings was 10.7% (95% CI: 4.4%–16.6%) and 55.4% (95% CI: 50.3%–60.1%), respectively. The VE against COVID-19 hospital death was 89.7% (95% CI: 77.7%–95.2%).

## Discussion

In this cohort study conducted among a socio-demographically diverse population in a large U.S. healthcare system, the bivalent (original and Omicron BA.4/BA.5) mRNA-1273 COVID-19 vaccine improved protection against a wide range of COVID-19 outcomes. mRNA-1273 BA.4/BA.5 bivalent vaccine effectiveness was high against hospitalization for COVID-19 disease and COVID-19 hospital death, even when compared to individuals who received ≥2 doses of monovalent mRNA vaccine; however, vaccine effectiveness was moderate against medically attended SARS-CoV-2 infection. The observed vaccine protection against hospitalization for COVID-19 was consistent across age (not estimated in the 6-17 years group), sex, race/ethnicity, IC status, history of SARS-CoV-2 infection, number of prior monovalent doses, and time since last monovalent dose. Additionally, the protection against hospitalization for COVID-19 was durable for at least 3 months of follow-up. These data from a real-world setting underscore the importance of remaining up-to-date with recommended COVID-19 vaccines, including receipt of a COVID-19 bivalent booster dose, to optimize protection, particularly against COVID-19 hospitalizations and death.

Initial estimates of vaccine effectiveness of bivalent mRNA vaccines have been generated using national outpatient and hospital-based surveillance networks in the United States. Early estimates from the VISION network of the VE and rVE (56% and 31–50%, respectively) against COVID-19 associated ED/UC visits during September–November 2022, when the BA.5 and other Omicron sublineages were the predominant SARS-CoV-2 variants in the United States, were similar to those in our study[13]. On the other hand, VISION's

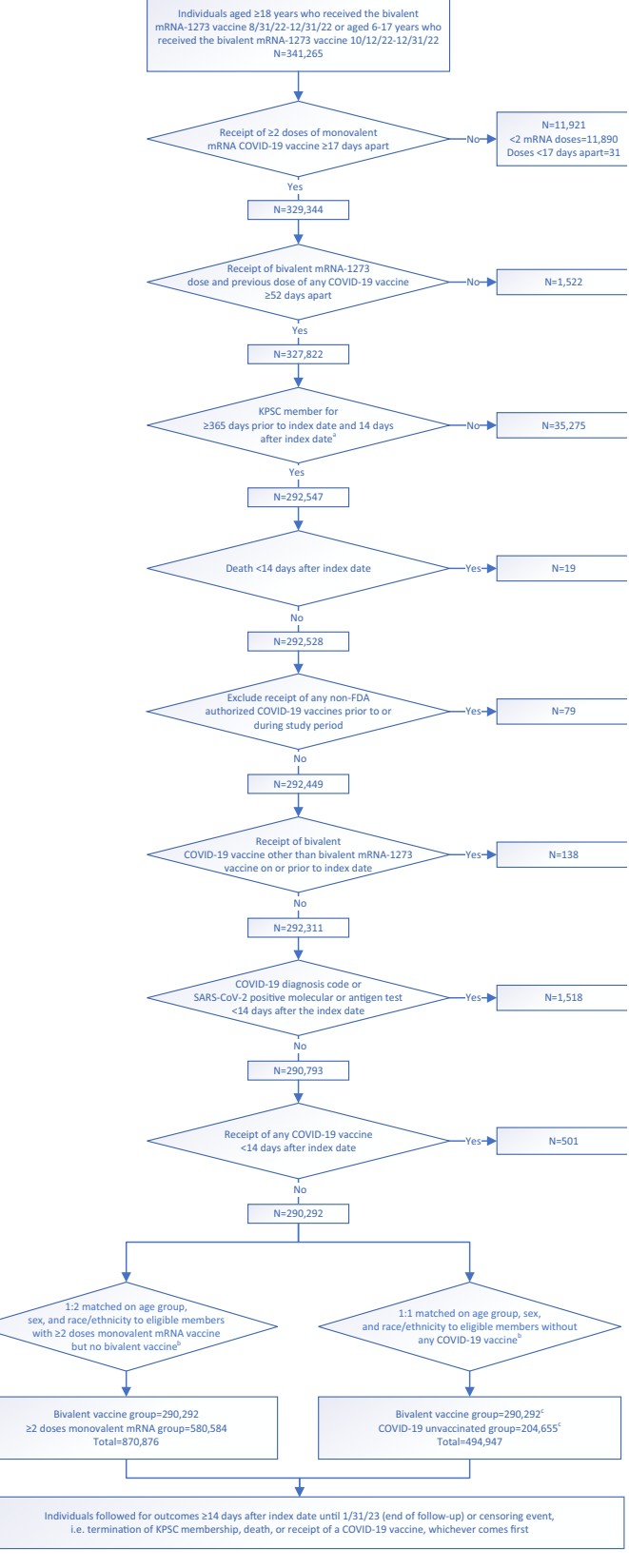

**Fig. 1 | Cohort selection for evaluating effectiveness of the bivalent (original and Omicron BA.4/BA.5) mRNA-1273 COVID-19 vaccine in Kaiser Permanente Southern California, 2022–2023 (N = 1,946,407).** Steps for selection of the bivalent mRNA-1273 cohort (n = 290,292), the ≥2 monovalent mRNA vaccine cohort (n = 580,584), and the COVID-19 unvaccinated cohort (n = 204,655) by inclusion and exclusion criteria in Kaiser Permanente Southern California (KPSC). [a]Index date for bivalent-vaccinated recipients is the date of their bivalent mRNA-1273 dose. For non-bivalent comparators, index date is assigned based on the bivalent dose date of their matched counterpart. [b]During the matching process, the following inclusion criteria were applied to non-bivalent comparators based on their assigned index date: no receipt of any bivalent COVID-19 vaccine on or prior to index date, no receipt of any COVID-19 vaccine <14 days after index date, no receipt of any non-U.S. Food and Drug Administration (FDA) authorized COVID-19 vaccines prior to or during study period, KPSC member for ≥365 days prior to index date and 14 days after index date, no death <14 days after index date, no COVID-19 diagnosis code or SARS-CoV-2 positive molecular or antigen test <14 days after index date, and any health care utilization or vaccination within two years prior to index date. [c]Not all bivalent-vaccinated individuals were able to match to a COVID-19 unvaccinated comparator. Unmatched bivalent-vaccinated individuals were kept in the analysis.

the bivalent booster against COVID-19 death in our study (89.7% and 82.7%, respectively) are higher than previous reports of bivalent VE and rVE against severe infection resulting in hospitalization or death[16,18].

Reduced VE of the mRNA monovalent vaccine series among IC adults has been found to be much more pronounced against infection with Omicron than with earlier variants[4], although VE against more severe COVID-19 outcomes was generally preserved. However, Britton and colleagues found that the VE of monovalent boosters against hospitalization with Omicron BA.2/BA.2.12.1/BA.4/BA.5 sublineages was markedly reduced among IC adults (32% ≥90 days after dose 3, and 43% ≥7 days after dose 4)[19]. In our study, we found that absolute VE of the bivalent booster against hospitalization for COVID-19 among a predominantly adult, IC population during periods of Omicron BA.5 and BQ.1 predominance was higher at 71.8%. The evidence supports the recommendation to boost the IC population with bivalent COVID-19 vaccine to lower the risk of hospitalization for COVID-19.

This study reports results that support early estimates of the effectiveness of bivalent COVID-19 boosters against an array of COVID-19 outcomes with data on durability and various subgroups. It has several strengths and limitations, with these limitations partly addressed through the strengths of the data source and analytic methods used in the study. First, there are various risk factors for infections and severe outcomes associated with testing and with vaccination that were unavailable in electronic health record (EHR) data. This could introduce bias, including mask-wearing, social distancing, and healthcare-seeking behavior. However, we attempted to mitigate potential bias by matching individuals and by adjusting for socio-demographic characteristics, prior healthcare utilization prior SARS-CoV-2 testing, and comorbidities in the models. In addition, we compared vaccinated and unvaccinated individuals at the same point in time, which balances community infection rates, exposure risk, and other secular impacts that might affect testing behaviors and risk of infection between vaccinated and unvaccinated participants. Furthermore, we estimated rVE, a comparison that should be less confounded by vaccination behavior, health care use, and social behavior differences than VE estimates with unvaccinated individuals as comparators. Although potential residual confounding could remain, it is unlikely to impact the conclusions of the study. Second, to reflect real-world conditions, we included in the analysis self-reported positive tests that were included in the EHR, but also required a COVID-19 diagnosis code in an encounter from 3 days before to 7 days after the test to ensure that the infection was medically attended. This likely reflected less severe disease. Third, although we attempted to account for a prior history of SARS-CoV-2 infection, it could be under-ascertained. Differential under-ascertainment could potentially generate bias. For example, some unvaccinated individuals could obtain some

estimates of VE and rVE against hospitalization (57% and 38%–45%, respectively[13]) were lower than our estimates (82.8% and 70.3%, respectively). It is possible that limiting our analysis to hospitalizations confirmed to have been for COVID-19 reduced the bias of VE estimates resulting from including hospitalizations for other reasons with incidental SARS-CoV-2 infection in the analysis[17]. The high VE and rVE of

**Table 1 | Comparison of baseline characteristics between individuals in the bivalent (original and Omicron BA.4/BA.5) mRNA-1273 COVID-19 vaccine cohort and the ≥2 monovalent mRNA vaccine cohort**

| | Bivalent vaccine group N = 290292 | ≥2 Monovalent mRNA vaccine group N = 580,584 | Total N = 870,876 | p value | Absolute standardized difference |
|---|---|---|---|---|---|
| Age at index date, years | | | | <0.01 | 0.05 |
| Mean (sd) | 58.67 (17.53) | 57.78 (18.43) | 58.07 (18.14) | | |
| Median | 62 | 61 | 61 | | |
| Q1, Q3 | 46, 72 | 46, 72 | 46, 72 | | |
| min, max | 6, 10 | 6, 11 | 6, 11 | | |
| Age at index date, years, n (%) | | | | N/A | N/A |
| 6–17 | 2715 (0.9%) | 5430 (0.9%) | 8145 (0.9%) | | |
| 18–44 | 639,53 (22.0%) | 127,906 (22.0%) | 191,859 (22.0%) | | |
| 45–64 | 96,293 (33.2%) | 192,586 (33.2%) | 288,879 (33.2%) | | |
| 65–74 | 73,258 (25.2%) | 146,516 (25.2%) | 219,774 (25.2%) | | |
| ≥75 | 54,073 (18.6%) | 108,146 (18.6%) | 162,219 (18.6%) | | |
| Sex, n (%) | | | | N/A | N/A |
| Female | 157,727 (54.3%) | 315,454 (54.3%) | 473,181 (54.3%) | | |
| Male | 132,565 (45.7%) | 265,130 (45.7%) | 397,695 (45.7%) | | |
| Race/ethnicity, n (%) | | | | N/A | N/A |
| Non-Hispanic White | 114,740 (39.5%) | 229,480 (39.5%) | 344,220 (39.5%) | | |
| Non-Hispanic Black | 23,517 (8.1%) | 47034 (8.1%) | 70551 (8.1%) | | |
| Hispanic | 82,547 (28.4%) | 165,094 (28.4%) | 247,641 (28.4%) | | |
| Non-Hispanic Asian | 50,129 (17.3%) | 100,258 (17.3%) | 150,387 (17.3%) | | |
| Other/unknown | 19,359 (6.7%) | 38718 (6.7%) | 58,077 (6.7%) | | |
| Body mass index[a], kg/m$^2$, n (%) | | | | <0.01 | 0.09 |
| <18.5 | 5233 (1.8%) | 10,502 (1.8%) | 15,735 (1.8%) | | |
| 18.5–<25 | 75,876 (26.1%) | 143,682 (24.7%) | 219,558 (25.2%) | | |
| 25–<30 | 89,285 (30.8%) | 176,138 (30.3%) | 265,423 (30.5%) | | |
| 30–<35 | 53,028 (18.3%) | 106,301 (18.3%) | 159,329 (18.3%) | | |
| 35–<40 | 24,218 (8.3%) | 46,561 (8.0%) | 70,779 (8.1%) | | |
| 40–<45 | 10,083 (3.5%) | 18,677 (3.2%) | 28,760 (3.3%) | | |
| ≥45 | 6432 (2.2%) | 11,188 (1.9%) | 17,620 (2.0%) | | |
| Unknown | 26,137 (9.0%) | 67,535 (11.6%) | 93,672 (10.8%) | | |
| Smoking[a], n (%) | | | | <0.01 | 0.09 |
| No | 216,879 (74.7%) | 419,564 (72.3%) | 63,6443 (73.1%) | | |
| Yes | 55,588 (19.1%) | 111,622 (19.2%) | 167,210 (19.2%) | | |
| Unknown | 17,825 (6.1%) | 49,398 (8.5%) | 67223 (7.7%) | | |
| Charlson comorbidity score[b], n (%) | | | | <0.01 | 0.06 |
| 0 | 162,456 (56.0%) | 342,290 (59.0%) | 504,746 (58.0%) | | |
| 1 | 50,534 (17.4%) | 92,596 (15.9%) | 143,130 (16.4%) | | |
| ≥2 | 77,302 (26.6%) | 145,698 (25.1%) | 223,000 (25.6%) | | |
| Frailty index[b] | | | | <0.01 | <0.01 |
| mean (sd) | 0.12 (0.03) | 0.12 (0.04) | 0.12 (0.03) | | |
| Median | 0.11 | 0.11 | 0.11 | | |
| Q1, Q3 | 0.10, 0.14 | 0.10, 0.14 | 0.10, 0.14 | | |
| Min, max | 0.04, 0.41 | 0.04, 0.43 | 0.04, 0.43 | | |
| Frailty index[b], n (%) | | | | <0.01 | 0.07 |
| Quartile 1 | 63,147 (21.8%) | 121,275 (20.9%) | 184,422 (21.2%) | | |
| Quartile 2 | 77,398 (26.7%) | 173,645 (29.9%) | 251,043 (28.8%) | | |
| Quartile 3 | 75,373 (26.0%) | 142,324 (24.5%) | 217,697 (25.0%) | | |
| Quartile 4, most frail | 74,374 (25.6%) | 143,340 (24.7%) | 217,714 (25.0%) | | |
| Chronic diseases[b], n (%) | | | | | |
| Kidney disease | 28,145 (9.7%) | 55,202 (9.5%) | 83,347 (9.6%) | <0.01 | 0.01 |
| Heart disease | 14,069 (4.8%) | 30,277 (5.2%) | 44,346 (5.1%) | <0.01 | 0.02 |
| Lung disease | 34,966 (12.0%) | 64,249 (11.1%) | 99,215 (11.4%) | <0.01 | 0.03 |
| Liver disease | 11,715 (4.0%) | 21,767 (3.7%) | 33,482 (3.8%) | <0.01 | 0.01 |
| Diabetes | 58,781 (20.2%) | 110,689 (19.1%) | 169,470 (19.5%) | <0.01 | 0.03 |

**Table 1 (continued) | Comparison of baseline characteristics between individuals in the bivalent (original and Omicron BA.4/BA.5) mRNA-1273 COVID-19 vaccine cohort and the ≥2 monovalent mRNA vaccine cohort**

| | Bivalent vaccine group<br>N = 290292 | ≥2 Monovalent mRNA vaccine group<br>N = 580,584 | Total<br>N = 870,876 | p value | Absolute standardized difference |
|---|---|---|---|---|---|
| Immunocompromised status, n (%) | | | | <0.01 | 0.04 |
| Yes | 12,338 (4.3%) | 19,991 (3.4%) | 32,329 (3.7%) | | |
| HIV/AIDS | 1894 | 1864 | 3758 | | |
| Leukemia, lymphoma, congenital and other immunodeficiencies, asplenia/hyposplenia | 5252 | 9283 | 14535 | | |
| Organ transplant | 1205 | 1767 | 2972 | | |
| Immunosuppressant medications | 6343 | 10,591 | 16,934 | | |
| Autoimmune conditions[b], n (%) | | | | <0.01 | 0.02 |
| Yes | 12,183 (4.2%) | 21,714 (3.7%) | 33,897 (3.9%) | | |
| Rheumatoid arthritis | 5206 | 9446 | 14,652 | | |
| Inflammatory bowel disease | 2063 | 3470 | 5533 | | |
| Psoriasis and psoriatic arthritis | 4641 | 8363 | 13,004 | | |
| Multiple sclerosis | 614 | 1042 | 1656 | | |
| Systemic lupus erythematosus | 807 | 1278 | 2085 | | |
| Pregnant at index date, n (%) | | | | <0.01 | 0.01 |
| Yes | 1200 (0.4%) | 2903 (0.5%) | 4103 (0.5%) | | |
| 1st trimester | 280 | 899 | 1179 | | |
| 2nd trimester | 462 | 1007 | 1469 | | |
| 3rd trimester | 458 | 997 | 1455 | | |
| History of SARS-CoV-2 infection[c], n (%) | | | | <0.01 | 0.07 |
| Yes | 69,405 (23.9%) | 155,360 (26.8%) | 224,765 (25.8%) | | |
| ≤180 days | 31,592 | 58,533 | 90,125 | | |
| 181–365 days | 20,310 | 50216 | 70,526 | | |
| >365 days | 17,503 | 46611 | 64,114 | | |
| History of SARS-CoV-2 molecular test[c], n (%) | 194,122 (66.9%) | 371,310 (64.0%) | 565,432 (64.9%) | <0.01 | 0.06 |
| Number of outpatient and virtual visits[b], n (%) | | | | <0.01 | 0.19 |
| 0 | 127,37 (4.4%) | 44,930 (7.7%) | 57,667 (6.6%) | | |
| 1–4 | 68,261 (23.5%) | 160,687 (27.7%) | 228,948 (26.3%) | | |
| 5–10 | 87,609 (30.2%) | 169,298 (29.2%) | 256,907 (29.5%) | | |
| ≥11 | 121,685 (41.9%) | 205,669 (35.4%) | 327,354 (37.6%) | | |
| Number of Emergency Department visits[b], n (%) | | | | <0.01 | 0.05 |
| 0 | 244,234 (84.1%) | 477,801 (82.3%) | 722,035 (82.9%) | | |
| 1 | 32,951 (11.4%) | 70,964 (12.2%) | 103,915 (11.9%) | | |
| ≥2 | 13107 (4.5%) | 31819 (5.5%) | 44,926 (5.2%) | | |
| Number of hospitalizations[b], n (%) | | | | <0.01 | 0.03 |
| 0 | 276,000 (95.1%) | 548,350 (94.4%) | 824,350 (94.7%) | | |
| 1 | 111,32 (3.8%) | 23,971 (4.1%) | 35,103 (4.0%) | | |
| ≥2 | 3160 (1.1%) | 8263 (1.4%) | 11,423 (1.3%) | | |
| Preventive care[b], n (%) | 251,979 (86.8%) | 446,555 (76.9%) | 698,534 (80.2%) | <0.01 | 0.26 |
| Medicaid, n (%) | 17,571 (6.1%) | 42,039 (7.2%) | 59,610 (6.8%) | <0.01 | 0.05 |
| Neighborhood median household income, n (%) | | | | <0.01 | 0.06 |
| <$40,000 | 8206 (2.8%) | 19,304 (3.3%) | 27,510 (3.2%) | | |
| $40,000–$59,999 | 44,119 (15.2%) | 95,199 (16.4%) | 139,318 (16.0%) | | |
| $60,000–$79,999 | 62,206 (21.4%) | 129,432 (22.3%) | 191,638 (22.0%) | | |
| ≥$80,000 | 175,589 (60.5%) | 336,116 (57.9%) | 511,705 (58.8%) | | |
| Unknown | 172 (0.1%) | 533 (0.1%) | 705 (0.1%) | | |
| Concomitant vaccination[d], n (%) | 50,729 (17.5%) | N/A | N/A | N/A | N/A |
| Antiviral therapy[e], n (%) | | | | <0.01 | 0.05 |
| Yes | 4094 (1.4%) | 4877 (0.8%) | 8971 (1.0%) | | |
| Nirmatrelvir/ritonavir | 4058 | 4832 | 8890 | | |
| Molnupiravir | 36 | 39 | 75 | | |
| Remdesivir | 3 | 8 | 11 | | |

**Table 1 (continued) | Comparison of baseline characteristics between individuals in the bivalent (original and Omicron BA.4/BA.5) mRNA-1273 COVID-19 vaccine cohort and the ≥2 monovalent mRNA vaccine cohort**

| | Bivalent vaccine group N = 290292 | ≥2 Monovalent mRNA vaccine group N = 580,584 | Total N = 870,876 | p value | Absolute standardized difference |
|---|---|---|---|---|---|
| Number of monovalent vaccines prior to index date[f], n (%) | | | | <0.01 | 0.64 |
| 2 doses | 11,493 (4.0%) | 135,437 (23.3%) | 146,930 (16.9%) | | |
| 3 doses | 144,052 (49.6%) | 287,077 (49.4%) | 431,129 (49.5%) | | |
| ≥4 doses | 134,747 (46.4%) | 158,070 (27.2%) | 292,817 (33.6%) | | |
| Time between latest monovalent vaccine and index date[f], days | | | | <0.01 | 0.40 |
| Mean (sd) | 263.24 (104.09) | 313.61 (144.69) | 296.82 (134.66) | | |
| Median | 260 | 312 | 302 | | |
| Q1, Q3 | 173, 343 | 189, 384 | 181, 364 | | |
| Min, max | 52, 716 | 1, 722 | 1, 722 | | |
| Time between latest monovalent vaccine and index date[f], n (%) | | | | <0.01 | 0.37 |
| ≤180 days | 83,060 (28.6%) | 134,392 (23.1%) | 217,452 (25.0%) | | |
| 181–365 days | 165,276 (56.9%) | 274,298 (47.2%) | 439,574 (50.5%) | | |
| >365 days | 41,956 (14.5%) | 171,894 (29.6%) | 213,850 (24.6%) | | |
| Number of monovalent mRNA vaccines prior to index date[f], n (%) | | | | N/A | N/A |
| 2 doses | 14,070 (4.8%) | 138,116 (23.8%) | 152,186 (17.5%) | | |
| 3 doses | 141,711 (48.8%) | 284,685 (49.0%) | 426,396 (49.0%) | | |
| ≥4 doses | 134,511 (46.3%) | 157,783 (27.2%) | 292,294 (33.6%) | | |
| Medical center area[g], n (%) | | | | <0.01 | 0.20 |
| Month of index date, n (%) | | | | N/A | N/A |
| September 2022 | 45,806 (15.8%) | 91,612 (15.8%) | 137,418 (15.8%) | | |
| October 2022 | 88,322 (30.4%) | 176,644 (30.4%) | 264,966 (30.4%) | | |
| November 2022 | 87,272 (30.1%) | 174,544 (30.1%) | 261,816 (30.1%) | | |
| December 2022 | 68,892 (23.7%) | 137,784 (23.7%) | 206,676 (23.7%) | | |

$\chi^2$ tests were used for categorical variables and two-sided, two-sample *t* tests were used for continuous variables.

*Min* minimum, *max* maximum, *N/A* not applicable, *Q* quartile, *sd* standard deviation.

[a]Defined in the 2 years prior to index date.

[b]Defined in the 1 year prior to index date.

[c]Defined based on all available medical records from 1 March 2020 to index date.

[d]Among subjects with concomitant vaccines received with the bivalent mRNA-1273 vaccine: influenza vaccine (89.6%), shingles vaccine (9.1%), pneumococcal vaccine (2.9%), Tdap (2.6%), and other vaccine (1.3%).

[e]Defined during follow-up.

[f]Defined based on all available vaccine records from 11 December 2020 to index date.

[g]Frequency and percent for the 19 medical center areas not shown.

protection from prior infections that were not documented in the EHR, resulting in underestimated VE. On the other hand, prior infections of vaccinated individuals might provide hybrid immunity from both infections and vaccinations, and if some of these prior infections were not accounted for in analyses, VE could be overestimated[20,21]. Fourth, misclassification of vaccination status is possible but unlikely to impact the results substantially. KPSC vaccination records capture all immunizations given within KPSC and are updated daily with California Immunization Registry data to which all providers are required by law to report COVID-19 vaccinations within 24 hours of administration. Fifth, we did not assess VE against asymptomatic infection. However, VE against a range of outcomes of varying severity (including medically attended infection, ED/UC visits, hospitalizations, and COVID-19 hospital deaths) provides estimates of protection from receipt of a bivalent booster that helps inform policy decisions. Sixth, the VE of COVID-19 vaccine against severe outcomes could potentially be more sustained given that cell-mediated immunity mechanisms appear to play a more significant role than humoral immunity in the prevention of severe outcomes[22,23]. However, with the wide confidence intervals for VE estimates and the replacement of BA.5 and BQ.1 by XBB sublineages, the durability of the bivalent vaccine (targeting Omicron BA.4/BA.5) against hospitalization due to BA.5 beyond 3 months is not as clear. Seventh, there were no cases of hospitalization for COVID-19 among exposed or unexposed individuals 6–17 years of age,

preventing estimation of rVE and VE in this age group. Finally, while our study was conducted during a period when the proportion of infections with highly immune-evasive XBB sublineages was growing, our data were insufficient to assess VE against XBB subvariants alone. Recent data suggest that short-term bivalent VE against COVID-19 disease with BA.5 and XBB sublineages may be similar[10].

This study found that vaccination with the bivalent (original and Omicron BA.4/BA.5) mRNA-1273 COVID-19 vaccine was associated with a lower risk of COVID-19 disease among recipients compared to those who received two or more doses of monovalent mRNA vaccines only or to those who were never vaccinated with a COVID-19 vaccine. These data suggest the potential benefit of receiving a COVID-19 booster dose directed against circulating SARS-CoV-2 variants, particularly for preventing severe outcomes associated with COVID-19. Additional studies are needed to assess the durability of protection afforded by BA.4/BA.5 bivalent vaccination, and other newly formulated vaccines against severe COVID-19 and their effectiveness against emerging SARS-CoV-2 variants, including XBB sublineages.

## Methods
### Study setting
KPSC is an integrated health system that provides health care services and insurance coverage to >4.8 million members with sociodemographic characteristics representative of the diverse

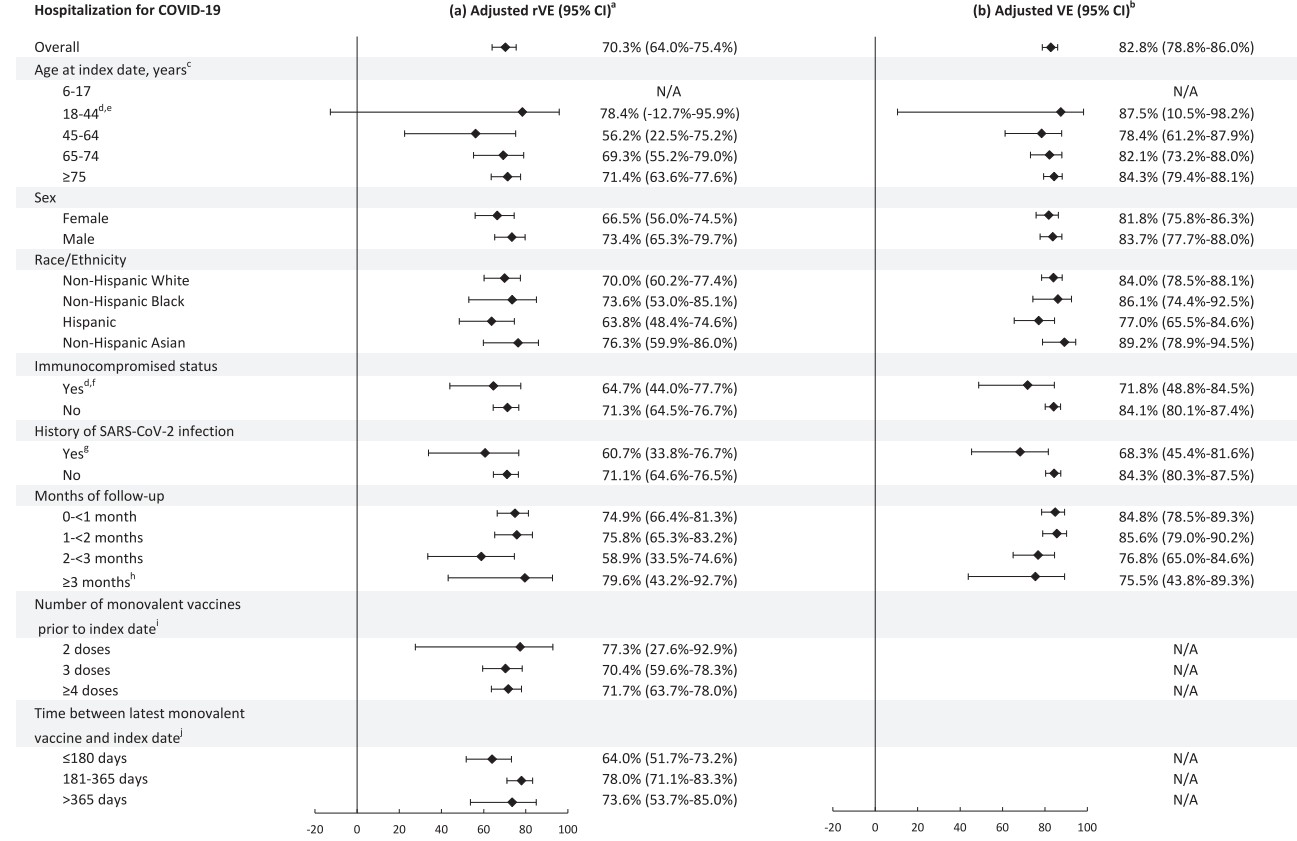

| Hospitalization for COVID-19 | (a) Adjusted rVE (95% CI)[a] | (b) Adjusted VE (95% CI)[b] |
|---|---|---|
| Overall | 70.3% (64.0%-75.4%) | 82.8% (78.8%-86.0%) |
| **Age at index date, years[c]** | | |
| 6-17 | N/A | N/A |
| 18-44[d,e] | 78.4% (-12.7%-95.9%) | 87.5% (10.5%-98.2%) |
| 45-64 | 56.2% (22.5%-75.2%) | 78.4% (61.2%-87.9%) |
| 65-74 | 69.3% (55.2%-79.0%) | 82.1% (73.2%-88.0%) |
| ≥75 | 71.4% (63.6%-77.6%) | 84.3% (79.4%-88.1%) |
| **Sex** | | |
| Female | 66.5% (56.0%-74.5%) | 81.8% (75.8%-86.3%) |
| Male | 73.4% (65.3%-79.7%) | 83.7% (77.7%-88.0%) |
| **Race/Ethnicity** | | |
| Non-Hispanic White | 70.0% (60.2%-77.4%) | 84.0% (78.5%-88.1%) |
| Non-Hispanic Black | 73.6% (53.0%-85.1%) | 86.1% (74.4%-92.5%) |
| Hispanic | 63.8% (48.4%-74.6%) | 77.0% (65.5%-84.6%) |
| Non-Hispanic Asian | 76.3% (59.9%-86.0%) | 89.2% (78.9%-94.5%) |
| **Immunocompromised status** | | |
| Yes[d,f] | 64.7% (44.0%-77.7%) | 71.8% (48.8%-84.5%) |
| No | 71.3% (64.5%-76.7%) | 84.1% (80.1%-87.4%) |
| **History of SARS-CoV-2 infection** | | |
| Yes[g] | 60.7% (33.8%-76.7%) | 68.3% (45.4%-81.6%) |
| No | 71.1% (64.6%-76.5%) | 84.3% (80.3%-87.5%) |
| **Months of follow-up** | | |
| 0-<1 month | 74.9% (66.4%-81.3%) | 84.8% (78.5%-89.3%) |
| 1-<2 months | 75.8% (65.3%-83.2%) | 85.6% (79.0%-90.2%) |
| 2-<3 months | 58.9% (33.5%-74.6%) | 76.8% (65.0%-84.6%) |
| ≥3 months[h] | 79.6% (43.2%-92.7%) | 75.5% (43.8%-89.3%) |
| **Number of monovalent vaccines prior to index date[i]** | | |
| 2 doses | 77.3% (27.6%-92.9%) | N/A |
| 3 doses | 70.4% (59.6%-78.3%) | N/A |
| ≥4 doses | 71.7% (63.7%-78.0%) | N/A |
| **Time between latest monovalent vaccine and index date[j]** | | |
| ≤180 days | 64.0% (51.7%-73.2%) | N/A |
| 181-365 days | 78.0% (71.1%-83.3%) | N/A |
| >365 days | 73.6% (53.7%-85.0%) | N/A |

**Fig. 2 | Relative vaccine effectiveness (rVE) and vaccine effectiveness (VE) of the bivalent (original and Omicron BA.4/BA.5) mRNA-1273 COVID-19 vaccine in preventing hospitalization for COVID-19, overall and by subgroups. a** Relative vaccine effectiveness (rVE) of the bivalent (original and Omicron BA.4/BA.5) mRNA-1273 COVID-19 vaccine, compared to the ≥2 monovalent mRNA vaccine group, and (**b**) Absolute vaccine effectiveness (VE) of the bivalent (original and Omicron BA.4/BA.5) mRNA-1273 COVID-19 vaccine, compared to the COVID-19 unvaccinated group and their 95% confidence intervals (CI) in preventing hospitalization for COVID-19, overall and by subgroups. Data are presented as rVE and VE and their 95% confidence intervals. Tabulated data and unadjusted estimates are available in Supplementary Tables 2 and 3. When the hazard ratio or its 95% CI was >1, the rVE or its 95% CI was transformed as ([1/hazard ratio]−1) × 100. [a]rVE models adjusted for covariates age group, sex, race/ethnicity, index date (in months), history of SARS-CoV-2 infection, number of outpatient and virtual visits, preventive care, number of monovalent vaccines prior to index date, time between latest monovalent vaccine and index date, and antiviral therapy. Medical center area removed from adjustment set due to lack of model convergence. [b]VE models

adjusted for covariates age group, sex, race/ethnicity, index date (in months), body mass index, smoking, Charlson comorbidity score, frailty index, kidney disease, lung disease, diabetes, immunocompromised status, history of SARS-CoV-2 infection, history of SARS-CoV-2 molecular test, number of outpatient and virtual visits, preventive care, Medicaid, and antiviral therapy. Neighborhood median household income and medical center area removed from adjustment set due to lack of model convergence. [c]Adjusted for continuous age (in years) in addition to covariates above. [d]Smoking removed from adjustment set in indicated VE models due to lack of model convergence. [e]Kidney disease, lung disease, and immunocompromised status removed from adjustment set in indicated VE models due to lack of model convergence. [f]Adjusted for immunocompromising sub-conditions in addition to covariates above. [g]Adjusted for time since prior SARS-CoV-2 infection in addition to covariates above. [h]Time between latest monovalent vaccine and index date removed from adjustment set due to lack of model convergence. [i]Number of monovalent vaccines prior to index date removed from adjustment set due to lack of model convergence.

Southern California population. Comprehensive EHRs capture details of patient care, including vaccinations, diagnoses, laboratory tests, procedures, and pharmacy records, from inpatient, ED, outpatient, and virtual care settings, with care received outside of the KPSC system captured through claims. In addition, vaccinations received outside of KPSC are imported daily from external sources, including the California Immunizations Registry (CAIR), Care Everywhere (system on the Epic EHR platform that allows different health care systems to exchange patients' medical information), claims (for example, retail pharmacies), and self-report by members (with valid documentation). The study was approved by the KPSC Institutional Review Board (#12758), which waived requirements for written informed consent and written Health Insurance Portability and Accountability Act authorization, as the use of EHRs for this observational study involved minimal risk. The study protocol was submitted to regulatory agencies prior to the conduct of the study[6].

### Study population

KPSC members were eligible for inclusion in the study if they were ≥6 years of age at the index date, had membership ≥12 months prior to the index date through 14 days after the index date, and had none of the exclusion criteria listed below. The exposure of interest was the bivalent (original and Omicron BA.4/BA.5) mRNA-1273 COVID-19 vaccine, as this study was conducted as part of a regulatory commitment from the manufacturer to multiple health authorities. The index date was the calendar date of receiving the first eligible bivalent mRNA-1273 COVID-19 vaccine for individuals in the exposed group. For unexposed individuals, the index date was assigned based on the index date of their exposed matched counterpart; these unexposed individuals had no receipt of bivalent vaccine prior to the index date. The exclusion criteria for both groups included receiving any bivalent COVID-19 vaccine other than bivalent mRNA-1273 vaccine on or prior to the index date, receipt of any non-FDA authorized COVID-19 vaccines prior to or during the follow-up, no health care utilization, and no vaccination

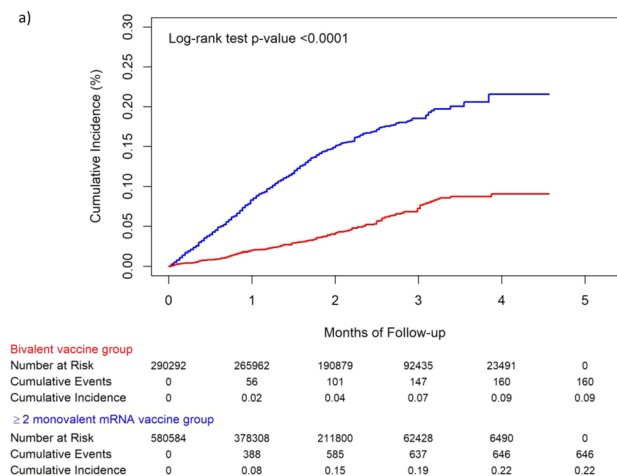
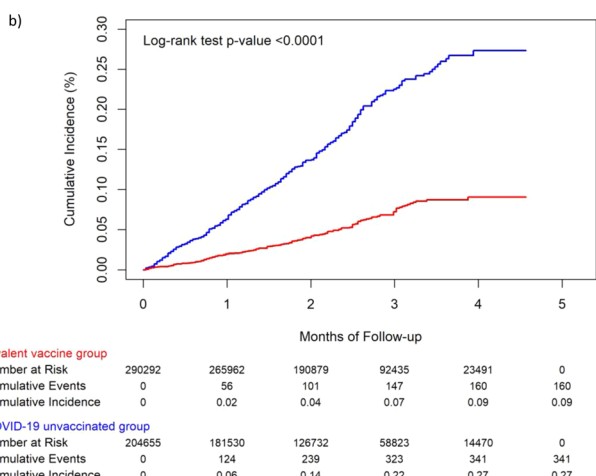

**Fig. 3 | Cumulative incidence of hospitalization for COVID-19 estimated by Kaplan Meier methods, comparing the bivalent mRNA-1273 cohort and the ≥2 monovalent mRNA vaccine cohort, and between the bivalent mRNA-1273 cohort and the COVID-19 unvaccinated cohort. a** Cumulative incidence of hospitalization for COVID-19 between individuals in the bivalent (original and Omicron BA.4/BA.5) mRNA-1273 COVID-19 vaccine cohort and the ≥2 monovalent mRNA vaccine cohort, and (**b**) cumulative incidence of hospitalization for COVID-19 between individuals in the bivalent (original and Omicron BA.4/BA.5) mRNA-1273 COVID-19 vaccine cohort and the COVID-19 unvaccinated cohort. The red line indicates the bivalent vaccine group, and the blue line indicates the comparison group. The difference in each comparison was tested by a log-rank test.

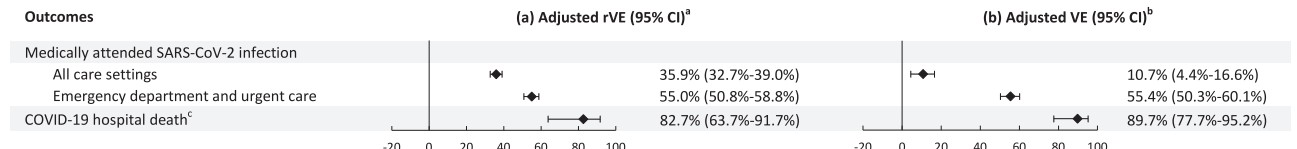

**Fig. 4 | Relative vaccine effectiveness (rVE) and vaccine effectiveness (VE) of the bivalent (original and Omicron BA.4/BA.5) mRNA-1273 COVID-19 vaccine in preventing medically attended SARS-CoV-2 infection in different settings and COVID-19 hospital death. a** Relative vaccine effectiveness (rVE) of the bivalent (original and Omicron BA.4/BA.5) mRNA-1273 COVID-19 vaccine, compared to the ≥2 monovalent mRNA vaccine group, and (**b**) Absolute vaccine effectiveness (VE) of the bivalent (original and Omicron BA.4/BA.5) mRNA-1273 COVID-19 vaccine, compared to the COVID-19 unvaccinated group and their 95% confidence intervals (CI) in preventing medically attended SARS-CoV-2 infection and COVID-19 hospital death. Data are presented as rVE and VE and their 95% confidence intervals. Tabulated data and unadjusted estimates are available in Supplementary Tables 4 and 5. [a]rVE models adjusted for covariates age group, sex, race/ethnicity, index date (in months), history of SARS-CoV-2 infection, number of outpatient and virtual visits, preventive care, number of monovalent vaccines prior to index date, time between latest monovalent vaccine and index date, and medical center area. [b]VE models adjusted for covariates age group, sex, race/ethnicity, index date (in months), BMI, smoking, Charlson comorbidity score, frailty index, kidney disease, lung disease, diabetes, immunocompromised status, history of SARS-CoV-2 infection, history of SARS-CoV-2 molecular test, number of outpatient and virtual visits, preventive care, Medicaid, neighborhood median household income, and medical center area. [c]Adjusted for antiviral therapy in addition to covariates above. Neighborhood median household income (VE models only) and medical center area (rVE and VE models) removed from adjustment set due to lack of model convergence.

within 2 years prior to the index date, receipt of any COVID-19 vaccine <14 days after the index date, death <14 days after the index date, or occurrence of a COVID-19 diagnosis code or a SARS-CoV-2 positive molecular or antigen test <14 days after the index date.

### Exposed cohort
Individuals were included in the exposed cohort if they received a dose of the bivalent mRNA-1273 COVID-19 vaccine (mRNA-1273.222 [original and Omicron BA.4/BA.5]) during the study accrual period (for ages ≥18 years: 8/31/2022-12/31/2022, and for ages 6–17 years: after FDA authorization on 10/12/2022-12/31/2022), received at least 2 monovalent mRNA COVID-19 vaccines prior to the bivalent dose, and did not receive any COVID-19 vaccine <52 days prior to the bivalent COVID-19 vaccine (recommended interval of ≥8 weeks, allowing a 4-day grace period).

### Unexposed cohorts
Two unexposed cohorts were used as comparators in the study.

1) ≥2 doses of monovalent mRNA COVID-19 vaccinated cohort (≥2 monovalent mRNA vaccine group): Individuals who had not received any bivalent dose but had received at least two doses of monovalent mRNA COVID-19 vaccine by the index date, were randomly selected and individually matched 2:1 to the bivalent exposed cohort.

2) COVID-19 unvaccinated group: Individuals who never received any COVID-19 vaccine dose by the index date were randomly selected and individually matched 1:1 to the bivalent exposed cohort.

The matching factors for both cohorts included age group (6–17 years, 18–44 years, 45–64 years, 65–74 years, and ≥75 years), sex (male, female), and race/ethnicity (Non-Hispanic White, Non-Hispanic Black, Hispanic, Non-Hispanic Asian, and Other/Unknown). Not all bivalent exposed individuals were matched with a COVID-19 unvaccinated individual due to a limited number of COVID-19 unvaccinated individuals, but all bivalent exposed individuals were retained in the analysis.

### Follow-up
The follow-up period ended on 01/31/2023. Each individual in the study was followed from 14 days after their index date till the end of the study follow-up period, membership end date, receipt of any COVID-19 vaccine, death, or outcome of interest, whichever came first.

## Outcome of interest

The primary outcome of interest was hospitalization for COVID-19, which included patients having a COVID-19 diagnosis code in the inpatient setting or a SARS-CoV-2 positive molecular or antigen test ≤7 days prior to or during the hospitalization stay. We ascertained the first occurrence of COVID-19 hospitalization ≥14 days after the index date. To ensure that hospitalization was *for* severe COVID-19 rather than coincident with SARS-CoV-2 infection, hospitalization for COVID-19 was further confirmed by (1) ≥1 documented oxygen saturation ($SpO_2$) of <90% during hospital stay for all patients or during a labor/delivery stay >2 days for pregnant patients or (2) manual chart review, as needed, performed by a physician investigator (B.K.A.) and trained chart abstractors to verify the presence of severe COVID-19 symptoms.

The secondary outcomes of interest included medically attended incident SARS-CoV-2 infection and COVID-19 hospital death. Medically attended SARS-CoV-2 infection was defined as infection resulting in seeking medical care in (1) all care settings: a SARS-CoV-2 positive molecular or antigen test, with a COVID-19 diagnosis code in the inpatient, emergency, outpatient, or virtual visit setting from 3 days before to 7 days after the test, or (2) ED/UC settings only: a SARS-CoV-2 positive molecular or antigen test, with a COVID-19 diagnosis code in the ED/UC settings from 3 days before to 7 days after the test. In both settings, the SARS-CoV-2 antigen test results could be self-reported. We ascertained the first occurrence of medically attended incident SARS-CoV-2 infection ≥14 days after the index date. COVID-19 hospital death occurred during a hospitalization for COVID-19.

## Covariates

Potential confounders were identified a priori based on the literature. Variables collected from EHRs at the index date included age, sex, race/ethnicity, socioeconomic status (Medicaid and neighborhood median household income), medical center area, and pregnancy status. Variables assessed prior to the index date included body mass index, smoking, Charlson comorbidity score, frailty index, chronic diseases, immunocompromised status, autoimmune conditions, health care visits (outpatient, virtual, ED, and inpatient), and preventive care (other vaccinations, screenings, and wellness visits). Race/ethnicity data (non-Hispanic White, non-Hispanic Black, Hispanic, non-Hispanic Asian, and other/unknown) were collected in the EHRs through self-report. Additional variables included history of SARS-CoV-2 infection and history of SARS-CoV-2 molecular test from 03/01/2020 to index date, and receipt of concomitant vaccine with the bivalent dose. For the analysis of the exposed cohort group versus the ≥2 monovalent mRNA vaccine group only, the number of monovalent COVID-19 vaccine doses prior to index date and time between the latest monovalent COVID-19 vaccine dose and index date were included. For analyses of hospitalization for COVID-19 and COVID-19 hospital death outcomes only, additional variables included were any antiviral therapy (nirmatrelvir/ritonavir, molnupiravir, or remdesivir) during follow-up (yes/no).

## Statistical analysis

We described attributes of the exposed and two unexposed cohorts. Categorical variables were compared using the $\chi^2$ test or Fisher's exact test, as appropriate, and continuous variables were compared using the two-sample $t$ test or Wilcoxon rank-sum test, as appropriate. Absolute standardized differences (ASD) were calculated to assess the balance of covariates; potential confounders were determined by ASD > 0.1 and were included in the adjusted models, along with matching variables (age, sex, and race/ethnicity), month of index date, and other select covariates based on scientific relevance.

We calculated overall incidence rates of hospitalization for COVID-19, medically attended SARS-CoV-2 infections, and COVID-19 hospital death for the exposed and the unexposed cohorts (number of incident events divided by person-years). The cumulative incidence of hospitalization for COVID-19 was estimated by the Kaplan–Meier method and compared between the exposed and unexposed cohorts by the log-rank test.

Unadjusted and adjusted hazard ratios (HR) and 95% confidence intervals (CIs) comparing hospitalization for COVID-19, medically attended SARS-CoV-2 infections, and COVID-19 hospital death between the exposed and the unexposed cohorts were estimated by Cox proportional hazards regression models without and with confounder adjustment. Relative vaccine effectiveness (rVE, %) comparing between the exposed and the ≥2 monovalent mRNA vaccine group and absolute vaccine effectiveness (VE, %) comparing between the exposed and the COVID-19 unvaccinated group, were calculated as $(1-HR) \times 100$ when HR was ≤1, and $([1/HR]-1) \times 100$ when HR was >1. We also assessed rVE and VE against the primary outcome, i.e., hospitalization for COVID-19, by age, sex, race/ethnicity, immunocompromised status, history of SARS-CoV-2 infection, time since bivalent mRNA-1273 COVID-19 vaccine, number of prior monovalent doses received (rVE only), and time since the last monovalent dose (rVE only). SAS 9.4 software (SAS Institute) was used for all analyses. Results were reported according to the Strengthening the Reporting of Observational Studies in Epidemiology (STROBE) checklist (Supplementary Table 6).

## Reporting summary

Further information on research design is available in the Nature Portfolio Reporting Summary linked to this article.

## Data availability

Individual-level data reported in this study involving human research participants are not publicly shared due to potentially identifying or sensitive patient information. Upon request to the corresponding author [H.F.T.], and subject to review and approval of an analysis proposal, KPSC may provide the deidentified aggregate-level data that support the findings of this study within 6 months. Anonymized data (deidentified data including participant data as applicable) that support the findings of this study may be made available from the investigative team in the following conditions: (1) agreement to collaborate with the study team on all publications, (2) provision of external funding for administrative and investigator time necessary for this collaboration, (3) demonstration that the external investigative team is qualified and has documented evidence of training for human subjects protections, and (4) agreement to abide by the terms outlined in data use agreements between institutions.

## Code availability

Standard epidemiological analyses were conducted using standard commands in SAS 9.4 (SAS Institute, Cary NC). The commands/code are available at https://doi.org/10.5281/zenodo.8274718[24].

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

## Acknowledgements

Medical writing and editorial assistance were provided by Joyce Lee, Ph.D., of MEDiSTRAVA in accordance with Good Publication Practice (GPP3) guidelines, funded by Moderna, Inc., and under the direction of the authors. The authors would like to acknowledge the following Kaiser Permanente Southern California staff: Maria Navarro, Elsa Olvera, Joy Gelfond, Jonathan Arguello, Kourtney Kottmann, Joanna Truong, Diana Romero, Samuel Payan, and Sierra Lewis for their contributions to manual chart reviews of the electronic health records. The authors would also like to acknowledge the contributions of Moderna staff: Groves Dixon, Ph.D., and Julie Vanas. The authors thank the patients of Kaiser Permanente for their partnership with us to improve their health. Their information, collected through our electronic health record systems, leads to findings that help us improve care for our members and can be shared with the larger community. This study was funded by Moderna, Inc. Employees of Moderna participated in the design and conduct of the study; collection, management, analysis, and interpretation of the data; preparation, review, or approval of the manuscript; or the decision to submit the manuscript for publication.

## Author contributions

H.F.T., B.K.A., L.S.S., K.J.B., C.K.Z., D.E., M.A.M., C.A.T., and L.Q. were involved in the study concept and design. H.F.T., B.K.A., L.S.S., J.E.T., Y.L., S.Q., G.S.L., K.J.B., J.H.K., A.F., H.S.T., R.B., C.K.Z., D.E., M.A.M., E.J.A., C.A.T., and L.Q. were involved in the acquisition, analysis, or interpretation of data. H.F.T. and B.K.A. drafted the manuscript. L.S.S., J.E.T., Y.L., S.Q., G.S.L., K.J.B., J.H.K., A.F., H.S.T., R.B., C.K.Z., D.E., M.A.M., E.J.A., C.A.T., and L.Q. critically revised the manuscript for important intellectual content. J.E.T., Y.L., S.Q., and L.Q. conducted the statistical analyses. H.F.T., G.S.L., H.S.T., M.A.M., and C.A.T. provided administrative, technical, or material support. H.F.T. and C.A.T. obtained funding. H.F.T., M.A.M., E.J.A., and C.A.T. provided supervision.

## Competing interests

H.F.T., B.K.A., L.S.S., J.E.T., Y.L., S.Q., G.S.L., J.H.K., A.F., H.S.T., R.B., and L.Q. are employees of Kaiser Permanente Southern California, which has been contracted by Moderna to conduct this study. K.J.B. is an adjunct investigator at Kaiser Permanente Southern California. C.K.Z., D.E., M.A.M., and E.J.A. are employees of and shareholders in Moderna, Inc. C.A.T. was an employee of and a shareholder in Moderna, Inc. at the time of analysis conception; C.A.T. is currently an employee of AstraZeneca. H.F.T. received funding from GlaxoSmithKline unrelated to this manuscript; H.F.T. also served on advisory boards for Janssen and Pfizer. B.K.A. received funding from GlaxoSmithKline, Dynavax, Genentech, and Pfizer unrelated to this manuscript. L.S.S. received funding from GlaxoSmithKline and Dynavax unrelated to this manuscript. J.E.T. received funding from Pfizer unrelated to this manuscript. Y.L. received funding from GlaxoSmithKline and Pfizer unrelated to this manuscript. S.Q. received funding from Dynavax unrelated to this manuscript. G.S.L.

received funding from GlaxoSmithKline unrelated to this manuscript. K.J.B. received funding from GlaxoSmithKline, Dynavax, Pfizer, and Gilead unrelated to this manuscript. J.H.K. received funding from GlaxoSmithKline unrelated to this manuscript. A.F. received funding from Pfizer, GlaxoSmithKline, and Gilead unrelated to this manuscript. H.S.T. received funding from GlaxoSmithKline, Pfizer, ALK, and Wellcome unrelated to this manuscript. R.B. received funding from GlaxoSmithKline unrelated to this manuscript. L.Q. received funding from GlaxoSmithKline and Dynavax unrelated to this manuscript.
