## [Peer Review File · Nature Communications]

mRNA-1273 bivalent (Original and Omicron) COVID-19 vaccine effectiveness against COVID-19 outcomes in the United StatesREVIEWER COMMENTS

Reviewer #1 (Remarks to the Author):

This manuscript assesses the vaccine effectiveness of the mRNA-1273 BA4/5 bivalent vaccine. It uses a cohort design and a large electronic health database which allows for adjustment for many factors.

This paper is overall well written, but I have a few areas which I would like some clarity.

Overall the rVE estimates seem very high at 70-80% this is compared to previous studies looking at the BA4/5 vaccine of around 50% (Link-Gelles et al MMWR 2023 Feb 3). Other studies looking at VE of the mRNA-1273 bivalent BA1 booster in England have also seen estimates of around 50% COVID-19 vaccine surveillance report: week 23 (publishing.service.gov.uk).

The high VE is addressed in the discussion making the point that it could be related to their ability to exclude incidental infections. I cannot see how this is the case when the outcome inclusion criteria includes using those with a ICD-10 coded admission only. I would like to know more about the inclusion of the COVID-19 diagnosis code outcomes – was this the primary reason for admission? Also what proportion of the total cases did not have a positive test associated with it? I also think the paragraph in the discussion relating to this (line 174) could also be made clearer “It is possible that limiting our analysis to hospitalizations confirmed to have been for COVID-19 may have reduced downward biasing of VE estimates observed when hospitalized individuals with incidental SARS-CoV-2 infection are included in the analysis”

My other question is related to the lack of waning at ≥ 3 as this has not been seen previously with other bivalent vaccines and I can't see this is addressed in the discussion. Other studies have seen faster waning after the BA1 boosters

https://assets.publishing.service.gov.uk/government/uploads/system/uploads/attachment_data/file/1161869/vaccine-surveillance-report-2023-week-23.pdf

Reviewer #2 (Remarks to the Author):

The manuscript, “Effectiveness of mRNA-1273 bivalent (Original and Omicron BA.4/BA.5) COVID-19 vaccine in preventing hospitalizations for COVID-19, medically attended SARS-CoV-2 infections, and hospital death in the United States” submitted by Tseng, et. al. uses a cohort study design and adds to the evidence of early effectiveness of bivalent vaccines against hospitalization, medically attended infection, and death. Below are specific comments:

1. As shown by others, bivalent mRNA vaccines provided additional protection compared with prior monovalent vaccination during the 3 months after bivalent dose. It seems like a missed opportunity that the analysis concludes its follow-up period on January 31, 2023, as it precludes assessment of potential immune evasion by XBB lineages, which only became predominant in the U.S. towards the end of January. Additionally, the follow-up period was too brief to demonstrate waning protection from bivalent mRNA COVID-19 vaccines, which has since been demonstrated. There have been several additional months of data since January 31, 2023, which could have provided more updated and relevant results. Could the authors please explain the rationale for the January 31, 2023 censor date?

2. The authors describe the primary outcome of interest as hospitalization for COVID-19 defined by either a COVID-19 diagnosis code in the inpatient setting or a SARS-CoV-2 molecular test ≤ 7 days prior to or during hospitalization. Additionally, COVID-19 patients were restricted to those who had at least one documented oxygen saturation $< 90\%$ during hospitalization to increase the likelihood that patients included in this analysis were hospitalized for, rather than with, COVID-19. What were SARS-CoV-2 testing practices for admitted patients during the period of this analysis? What proportion of COVID-19 patients did not have a COVID-19 diagnosis code, but did have a positive SARS-CoV-2 test?

3. Although patients aged 6-17 years were eligible for inclusion, Table 1 shows that only 0.9% of bivalent exposed and unexposed patients in this age group were included in the analysis. Furthermore, Supplementary Table 2 shows no cases of hospitalized COVID-19 in either exposed or unexposed groups, preventing rVE or VE estimates for patients aged 6–17 years. As such, the conclusions in this analysis are based on adults only. This point should be clearly stated in the Discussion (e.g., line 160). Consider restricting this analysis to adults aged at least 18 years.

4. The months of follow-up in Figure 2 includes a row for “≥3 months” follow-up time. What was the upper bound of the range of the follow-up time in this category? It would be more precise to provide time bounds for this category to avoid giving the impression of sustained protection well beyond 3 months, especially since median follow-up time was 2.46 months for bivalent exposed patients and 1.45 months for unexposed patients. Additionally, line 162 suggests durable protection against hospitalization for COVID-19 for “3 or more” months of follow-up, but waning effectiveness of bivalent mRNA vaccines against hospitalization has been demonstrated (Link-Gelles R, et. al. *MMWR Morb Mortal Wkly Rep* 2023;72:579-588 and Lin DY, Xu Y, Gu Y, Zeng D, Sunny SK, Moore Z. Durability of Bivalent Boosters against Omicron Subvariants. *N Engl J Med*. 2023 May 11;388(19):1818-1820). Figure 2 also shows decreases in point estimates for rVE and VE ≥3 months after bivalent dose, although sample size in this group is small, confidence intervals are wide and overlap with earlier periods. The insufficient follow-up time to meaningfully assess durability of protection should be listed as a limitation and earlier statements of durable protection for 3 or more months should have time bounds.

5. The sentence in lines 152-155 suggests that this analysis provided evidence of bivalent vaccine effectiveness against a range of emerging Omicron sub-lineages despite evidence of immunological escape. However, the only lineage that was predominant during the period of this analysis (other than BA.4/5) was BQ.1, which is derived from BA.5. Omicron lineages BA.2.75.2 and XBB.1 included in the reference provided (ref #7) were not assessed in this analysis. Greater specificity in this statement is needed.

6. Reference #19 (DeCuir, et. al.) has been used inaccurately as supporting evidence of bivalent mRNA vaccine protection against death whereas the report describes monovalent, not bivalent, mRNA COVID-19 vaccine effectiveness against these severe outcomes.

Reviewer #3 (Remarks to the Author):

Thank you for the opportunity to review this important and timely analysis. This is an important study that provides estimates for the effectiveness of the bivalent mRNA-1273 vaccine, combining ancestral virus and omicron BA.4 and BA.5 strains, against COVID-19 hospitalization, symptomatic infections that required medical attention, and COVID-19 deaths. The study is based on a large cohort of individuals that are part of a sizable insurance network in South California. The results showed high effectiveness of the bivalent vaccine against hospitalization that lasts for at least 3 months as well as high effectiveness against COVID-19 deaths, but only modest to moderate effectiveness against symptomatic infection. The study findings are of global interest, inform decisions about vaccination, and are in agreement with a substantial body of evidence demonstrating effectiveness of vaccines in mitigating COVID-19 severe outcomes. I have suggested few revisions in the spirit of providing further clarifications of the methods used in this study.

1) It was a bit confusing to me to have relative VE reported for the first cohort analysis but absolute VE reported for the second. Why not report relative VE and absolute VE for both analyses? I encourage authors to include the rationale behind this choice in the statistical analysis section.

2) I found it unclear as to why the authors chose to include unmatched bivalent individuals in the analysis. This decision appears to contradict the purpose of matching, which is to balance observed confounders between the exposed and unexposed groups and derive an unbiased estimate of vaccine effectiveness. Those who received the bivalent vaccine that could not be matched in the

numerator of the hazard ratio may have a very different risk from that of the comparator group in the denominator. In the analysis comparing the outcomes between the bivalent cohort and those unvaccinated, there were 85,637 unmatched individuals which comprise about a third of the exposed group. I am not sure if the same strategy was used in the first analysis comparing the bivalent group to the monovalent group, and it will be great if the authors mention this point explicitly. Regardless, I strongly recommend that authors include as a main analysis estimates of VE based only on matched individuals in both cohorts. This in my view will provide more robust VE results.

3) It was not clear if authors used exact matching or propensity score matching. I think it will be useful to clarify this point.

4) I encourage authors to include explicitly how they defined prior infection was it a 90-day cutoff?

5) I encourage authors if the sample size allows to consider matching by any or a combination of the Charlson comorbidity index, number of vaccine doses, and prior infection status.

6) History of prior infection was included in the model as yes/no. I encourage authors if the sample size allows to consider including the variable with the number of days since infection as per Table 1. It will be interesting to note differential effectiveness for the vaccine based on time since prior infection

7) It may be useful to add the Kaplan-Meier for secondary outcomes and including the number of events in the supplementary material.

8) The authors may wish to consider including the STROBE checklist as a supplementary file. This addition would ensure that all essential items required for reporting have been appropriately included and adhered to in the study.

REVIEWER COMMENTS

Reviewer #1 (Remarks to the Author):

This manuscript assesses the vaccine effectiveness of the mRNA-1273 BA4/5 bivalent vaccine. It uses a cohort design and a large electronic health database which allows for adjustment for many factors.

This paper is overall well written, but I have a few areas which I would like some clarity.

Overall the rVE estimates seem very high at 70-80% this is compared to previous studies looking at the BA4/5 vaccine of around 50% (Link-Gelles et al MMWR 2023 Feb 3). Other studies looking at VE of the mRNA-1273 bivalent BA1 booster in England have also seen estimates of around 50% COVID-19 vaccine surveillance report: week 23 (publishing.service.gov.uk).

The high VE is addressed in the discussion making the point that it could be related to their ability to exclude incidental infections. I cannot see how this is the case when the outcome inclusion criteria includes using those with a ICD-10 coded admission only. I would like to know more about the inclusion of the COVID-19 diagnosis code outcomes – was this the primary reason for admission? Also what proportion of the total cases did not have a positive test associated with it?

Response: Thanks for the comments. Those who were identified by a COVID-19 diagnosis code in the inpatient setting were considered “potential” COVID-19 hospitalization cases. To ensure that hospitalization was for severe COVID-19 rather than coincident with SARS-CoV-2 infection, hospitalization for COVID-19 was further confirmed by (1) ≥ 1 documented oxygen saturation (SpO_2) of $<90\%$ during a hospital stay for all patients or during a labor/delivery stay >2 days for pregnant patients or (2) manual chart review performed by a physician investigator (B.K.A.) and trained chart abstractors to verify the presence of severe COVID-19 symptoms. The same criteria were also applied to potential COVID-19 hospitalization cases that were identified by a positive molecular test or antigen test during or in the week prior to hospitalization.

Among 1,695 potential hospitalized patients, 387 (22.8%) were confirmed by the SpO_2 criteria; of the remaining 1308 cases that were chart reviewed, 760 (58.1%) were confirmed as hospitalized for COVID-19. In a previous analysis, we validated that the positive predictive value of a single SpO_2 value $<90\%$ in identifying hospitalization for COVID-19 was 95.7% (179/187).

Among the 1,147 confirmed hospitalizations in the three exposure groups, 22.3% did not have an associated positive test and only had a COVID-19 diagnosis code (see table below). For hospitalization records from claims (non-KPSC hospitals), the lab results were not available to us.

Hospitalizations for COVID-19 among the three exposure groups.

	Bivalent vaccine group (N=160)	≥2 monovalent mRNA vaccine group (N=646)	COVID-19 unvaccinated group (N=341)	Total (N=1147)
Hospitalization for COVID-19, n (%)				
COVID-19 diagnosis code and positive lab	107 (66.9%)	434 (67.2%)	205 (60.1%)	746 (65.0%)
COVID-19 diagnosis code only	27 (16.9%)	129 (20.0%)	100 (29.3%)	256 (22.3%)
Positive lab only	26 (16.3%)	83 (12.8%)	36 (10.6%)	145 (12.6%)

The referenced UK report notes that there are likely still some incidental admissions in their hospitalization data. This may contribute to lower initial VE estimates compared to other studies in which additional measures are taken to minimize the inclusion of persons with incidental SARS-CoV-2 infection hospitalized for unrelated reasons;¹ it may also increase apparent waning.²

Furthermore, in the reference (Link-Gelles et al MMWR 2023 Feb 3) noted by the reviewer, Link-Gelles and colleagues reported that VE against non-medically attended symptomatic infection identified by testing done in a pharmacy network would be expected to be much lower than VE against hospitalization.³ In that study, the rVE against symptomatic test-positive BA.5-related infection (less severe compared to hospitalization) ranged from 37-52%, depending on age group. Our rVE against medically attended SARS-CoV-2 infection (ED/Urgent care) was 45%, which was comparable to the referenced study.

In addition, there is a study among older adults (≥65 years of age) that reported early VE against hospitalization that is similar to estimates observed in this study (aVE 84%, rVE 73%).⁴

I also think the paragraph in the discussion relating to this (line 174) could also be made clearer “It is possible that limiting our analysis to hospitalizations confirmed to have been for COVID-19 may have reduced downward biasing of VE estimates observed when hospitalized individuals with incidental SARS-CoV-2 infection are included in the analysis”

Response: Thanks for the comment. We have rephrased the sentence as “It is possible that limiting our analysis to hospitalizations confirmed to have been for COVID-19 reduced the bias of VE estimates resulting from including hospitalizations for other reasons with incidental SARS-CoV-2 infection in the analysis.¹⁷” (Lines 177-180).

My other question is related to the lack of waning at =>3 as this has not been seen previously with other bivalent vaccines and I can't see this is addressed in the discussion. Other studies have seen faster waning after the BA1 boosters

https://assets.publishing.service.gov.uk/government/uploads/system/uploads/attachment_data/fi

Response: Thanks for the comment. The VE of COVID-19 vaccines against severe outcomes could potentially be more sustained given that cell-mediated immunity mechanisms appear to play a more significant role than humoral immunity in prevention of severe outcomes.^{5,6} However, with the wide confidence intervals for VE estimates and the replacement of BA.5 and BQ.1 by XBB sublineages, the durability of the bivalent vaccine (targeting Omicron BA.4/BA.5) against hospitalization due to BA.5 beyond 3 months is not as clear. We have included this in the Discussion section (Lines 228-233).

Reviewer #2 (Remarks to the Author):

The manuscript, “Effectiveness of mRNA-1273 bivalent (Original and Omicron BA.4/BA.5) COVID-19 vaccine in preventing hospitalizations for COVID-19, medically attended SARS-CoV-2 infections, and hospital death in the United States” submitted by Tseng, et. al. uses a cohort study design and adds to the evidence of early effectiveness of bivalent vaccines against hospitalization, medically attended infection, and death. Below are specific comments:

1. As shown by others, bivalent mRNA vaccines provided additional protection compared with prior monovalent vaccination during the 3 months after bivalent dose. It seems like a missed opportunity that the analysis concludes its follow-up period on January 31, 2023, as it precludes assessment of potential immune evasion by XBB lineages, which only became predominant in the U.S. towards the end of January. Additionally, the follow-up period was too brief to demonstrate waning protection from bivalent mRNA COVID-19 vaccines, which has since been demonstrated. There have been several additional months of data since January 31, 2023, which could have provided more updated and relevant results. Could the authors please explain the rationale for the January 31, 2023 censor date?

Response: Thanks for the comment. The follow-up ended on 1/31/2023. This analysis plan was pre-specified as a regulatory commitment. With an additional 2-3 months for data settling, followed by chart review, statistical analysis, and manuscript drafting and reviewing among co-authors, we were able to submit the manuscript on May 26, 2023. This timeline was developed to ensure timely publication, although at the cost of a shorter follow-up.

2. The authors describe the primary outcome of interest as hospitalization for COVID-19 defined by either a COVID-19 diagnosis code in the inpatient setting or a SARS-CoV-2 molecular test ≤ 7 days prior to or during hospitalization. Additionally, COVID-19 patients were restricted to those who had at least one documented oxygen saturation $< 90\%$ during hospitalization to increase the likelihood that patients included in this analysis were hospitalized for, rather than with, COVID-19. What were SARS-CoV-2 testing practices for admitted patients during the

period of this analysis? What proportion of COVID-19 patients did not have a COVID-19 diagnosis code, but did have a positive SARS-CoV-2 test?

Response: Thanks for the comment. A patient with a SARS-CoV-2 positive molecular or antigen test ≤ 7 days prior to or during the hospitalization stay would be included as a “potential” hospitalized case for the study. The hospitalized cases for COVID-19 were then also confirmed by (1) ≥ 1 documented oxygen saturation (SpO_2) of $< 90\%$ during hospital stay or during a labor/delivery stay > 2 days for pregnant patients or (2) manual chart review performed by a physician investigator (B.K.A.) and trained chart abstractors to verify the presence of severe COVID-19 symptoms. The same criteria were also applied to potential COVID-19 hospitalization cases that were identified by a COVID-19 diagnosis code in the inpatient setting.

Among 1,695 potential hospitalized patients, 387/1695 (22.8%) were confirmed by the SpO_2 criteria; of the remaining 1308 cases that were chart reviewed, 760 (58.1%) were confirmed as hospitalized for COVID-19. In a previous analysis, we validated that the positive predictive value of a single $SpO_2 < 90\%$ in identifying hospitalization for COVID-19 was 95.7% (179/187).

Among the 1,147 confirmed hospitalizations in the three exposure groups, 12.6% did not have an associated COVID-19 diagnosis and only had a positive SARS-CoV-2 test (see table in response to Reviewer #1). During the study period, every admitted patient had to be tested for COVID-19 in KPSC hospitals. For a small number of hospital cases from claims (i.e. non-KPSC hospitals), automated lab data was not readily available in the KPSC medical record. However, this information was nearly always documented in the claim’s documentation, Care Everywhere (a medical record exchange between KPSC and non-KPSC facilities for care provided to KPSC members), or routine post-hospital discharge visits, and was reviewed as part of manual chart review. In addition, in nearly all cases, sufficient clinical documentation was available from these sources to distinguish hospitalizations for COVID-19 from hospitalizations with coincident SARS-CoV-2 infection. In rare occasions when the information is insufficient, we would not include the patients as hospitalizations for COVID-19.

3. Although patients aged 6-17 years were eligible for inclusion, Table 1 shows that only 0.9% of bivalent exposed and unexposed patients in this age group were included in the analysis. Furthermore, Supplementary Table 2 shows no cases of hospitalized COVID-19 in either exposed or unexposed groups, preventing rVE or VE estimates for patients aged 6–17 years. As such, the conclusions in this analysis are based on adults only. This point should be clearly stated in the Discussion (e.g., line 160). Consider restricting this analysis to adults aged at least 18 years.

Response: Thanks for the comment. Indeed, in Figure 2, the rVE and VE were not estimated in the population aged 6-17 years. We have revised Line 162 as “The observed vaccine protection against hospitalization for COVID-19 was consistent across age (not estimated in the 6-17 years group), sex, race/ethnicity, IC status, history of SARS-CoV-2 infection, number of prior monovalent doses, and time since last monovalent dose.” Also, we have added the following sentence in the limitation section of the Discussion: “Seventh, there were no cases of hospitalization for COVID-19 among

exposed or unexposed individuals 6-17 years of age, preventing estimation of rVE and VE in this age group.” (Lines 234-235).

4. The months of follow-up in Figure 2 includes a row for “≥3 months” follow-up time. What was the upper bound of the range of the follow-up time in this category? It would be more precise to provide time bounds for this category to avoid giving the impression of sustained protection well beyond 3 months, especially since median follow-up time was 2.46 months for bivalent exposed patients and 1.45 months for unexposed patients. Additionally, line 162 suggests durable protection against hospitalization for COVID-19 for “3 or more” months of follow-up, but waning effectiveness of bivalent mRNA vaccines against hospitalization has been demonstrated (Link-Gelles R, et. al. MMWR Morb Mortal Wkly Rep 2023;72:579-588 and Lin DY, Xu Y, Gu Y, Zeng D, Sunny SK, Moore Z. Durability of Bivalent Boosters against Omicron Subvariants. N Engl J Med. 2023 May 11;388(19):1818-1820). Figure 2 also shows decreases in point estimates for rVE and VE ≥3 months after bivalent dose, although sample size in this group is small, confidence intervals are wide and overlap with earlier periods. The insufficient follow-up time to meaningfully assess durability of protection should be listed as a limitation and earlier statements of durable protection for 3 or more months should have time bounds.

Response: Thanks for the comment. We have added the maximum follow-up time (4.57 months) to the Results section (Lines 82, 85, 91). We have also added a footnote indicating the maximum follow-up time for the relevant figures and tables. We have revised Line 164-165 as “Additionally, the protection against hospitalization for COVID-19 was durable for at least 3 months of follow-up.” Finally, we have added this as a limitation in the Discussion: “Sixth, the VE of COVID-19 vaccine against severe outcomes could potentially be more sustained given that cell-mediated immunity mechanisms appear to play a more significant role than humoral immunity in prevention of severe outcomes.^{22,23} However, with the wide confidence intervals for VE estimates and the replacement of BA.5 and BQ.1 by XBB sublineages, the durability of the bivalent vaccine (targeting Omicron BA.4/BA.5) against hospitalization due to BA.5 beyond 3 months is not as clear.” (Lines 228-233).

5. The sentence in lines 152-155 suggests that this analysis provided evidence of bivalent vaccine effectiveness against a range of emerging Omicron sub-lineages despite evidence of immunological escape. However, the only lineage that was predominant during the period of this analysis (other than BA.4/5) was BQ.1, which is derived from BA.5. Omicron lineages BA.2.75.2 and XBB.1 included in the reference provided (ref #7) were not assessed in this analysis. Greater specificity in this statement is needed.

Response: Thanks for the comment. The sentences in Lines 155-157 were deleted.

6. Reference #19 (DeCuir, et. al.) has been used inaccurately as supporting evidence of bivalent mRNA vaccine protection against death whereas the report describes monovalent, not bivalent, mRNA COVID-19 vaccine effectiveness against these severe outcomes.

Response: Thanks for the comment. We have deleted this reference and renumbered the subsequent references.

Reviewer #3 (Remarks to the Author):

Thank you for the opportunity to review this important and timely analysis. This is an important study that provides estimates for the effectiveness of the bivalent mRNA-1273 vaccine, combining ancestral virus and omicron BA.4 and BA.5 strains, against COVID-19 hospitalization, symptomatic infections that required medical attention, and COVID-19 deaths. The study is based on a large cohort of individuals that are part of a sizable insurance network in South California. The results showed high effectiveness of the bivalent vaccine against hospitalization that lasts for at least 3 months as well as high effectiveness against COVID-19 deaths, but only modest to moderate effectiveness against symptomatic infection. The study findings are of global interest, inform decisions about vaccination, and are in agreement with a substantial body of evidence demonstrating effectiveness of vaccines in mitigating COVID-19 severe outcomes. I have suggested few revisions in the spirit of providing further clarifications of the methods used in this study.

1) It was a bit confusing to me to have relative VE reported for the first cohort analysis but absolute VE reported for the second. Why not report relative VE and absolute VE for both analyses? I encourage authors to include the rationale behind this choice in the statistical analysis section.

Response: Thanks for the comment. The two analyses used different comparison groups. Relative VE (rVE) is the VE comparing between the bivalent vaccinated group and another vaccinated group (at least two monovalent mRNA vaccines, but no bivalent vaccine), while absolute VE (aVE) is the VE comparing between the bivalent vaccinated group and the unvaccinated group (COVID-19 vaccine naïve group).

2) I found it unclear as to why the authors chose to include unmatched bivalent individuals in the analysis. This decision appears to contradict the purpose of matching, which is to balance observed confounders between the exposed and unexposed groups and derive an unbiased estimate of vaccine effectiveness. Those who received the bivalent vaccine that could not be matched in the numerator of the hazard ratio may have a very different risk from that of the comparator group in the denominator. In the analysis comparing the outcomes between the bivalent cohort and those unvaccinated, there were 85,637 unmatched individuals which comprise about a third of the exposed group. I am not sure if the same strategy was used in the first analysis comparing the bivalent group to the monovalent group, and it will be great if the authors mention this point explicitly. Regardless, I strongly recommend that authors include as a main analysis estimates of VE based only on matched individuals in both cohorts. This in my view will provide more robust VE results.

Response: Thanks for the comment. In the rVE analysis (comparing the bivalent vaccinated cohort and monovalent vaccinated cohort), which is the primary analysis of the study, all individuals in the bivalent group were matched with monovalent vaccinated individuals. In the aVE analysis (comparing the bivalent vaccinated cohort and the unvaccinated cohort), not all bivalent vaccinated individuals were matched, due to a limited number of unvaccinated individuals, particularly among the older age groups. The unmatched bivalent vaccinated individuals were included in the analysis to increase the precision of the estimates. While the two cohorts were not 100% matched on age, sex, and race/ethnicity, we adjusted for these variables (forced in) in the multivariable

analyses. By increasing the sample size in the bivalent group (since the incidence rate of hospitalization in this group was expected to be lower), we were able to increase the precision of estimates without introducing bias from confounding.

3) It was not clear if authors used exact matching or propensity score matching. I think it will be useful to clarify this point.

Response: Thanks for the comment. We used exact matching as stated in the Methods section: “The matching factors for both cohorts included age group (6-17 years, 18-44 years, 45-64 years, 65-74 years, and ≥ 75 years), sex (male, female), and race/ethnicity (Non-Hispanic White, Non-Hispanic Black, Hispanic, Non-Hispanic Asian, and Other/Unknown).” (Lines 376-378).

4) I encourage authors to include explicitly how they defined prior infection was it a 90-day cutoff?

Response: Thanks for the comment. History of prior SARS-CoV-2 infection was defined based on all available medical records from March 1, 2020 to the index date. This was included in the Methods section under Covariates and in footnotes in the tables.

5) I encourage authors if the sample size allows to consider matching by any or a combination of the Charlson comorbidity index, number of vaccine doses, and prior infection status.

Response: Thanks for the comment. Increasing the number of matching variables increases the difficulty of matching due to a reduction in sample size. As a result, adjustment for additional covariates was utilized in our regression models. Absolute standardized differences were used to identify covariates for potential adjustment and variables with an absolute standardized difference exceeding 0.1 were included in the multivariable model to control for potential differences. Also, stratified analyses by number of monovalent vaccine doses and by history of SARS-CoV-2 infection were also conducted.

6) History of prior infection was included in the model as yes/no. I encourage authors if the sample size allows to consider including the variable with the number of days since infection as per Table 1. It will be interesting to note differential effectiveness for the vaccine based on time since prior infection.

Response: Thanks for the comments. We attempted stratified analysis by time since prior infection, but the models failed to converge due to small sample size.

7) It may be useful to add the Kaplan-Meier for secondary outcomes and including the number of events in the supplementary material.

Response: Thanks for the comment. We propose not to include the K-M curves in the manuscript as they were for secondary outcomes and were unadjusted risks. We are providing them in the figures below.

Figure 1. Cumulative incidence of medically attended SARS-CoV-2 infection in all settings among the bivalent (original and Omicron BA.4/BA.5) mRNA-1273 COVID-19 vaccine cohort, ≥ 2 monovalent mRNA vaccine cohort, and COVID-19 unvaccinated cohort.

	Months of Follow-up					
Bivalent vaccine group						
Number at Risk	290292	265012	189286	91049	23006	0
Cumulative Events	0	1041	2103	2944	3193	3221
Cumulative Incidence	0	0.37	0.83	1.42	1.84	2.17
≥ 2 monovalent mRNA vaccine group						
Number at Risk	580584	376130	209182	61425	6381	0
Cumulative Events	0	2687	4348	4883	4963	4964
Cumulative Incidence	0	0.57	1.13	1.52	1.79	1.81
COVID-19 unvaccinated group						
Number at Risk	204655	181016	126014	58294	14342	0
Cumulative Events	0	649	1182	1510	1574	1578
Cumulative Incidence	0	0.33	0.67	1.02	1.2	1.24

Figure 2. Cumulative incidence of medically attended SARS-CoV-2 infection in emergency department/urgent care settings among the bivalent (original and Omicron BA.4/BA.5) mRNA-

1273 COVID-19 vaccine cohort, ≥ 2 monovalent mRNA vaccine cohort, and COVID-19 unvaccinated cohort.

Months of Follow-up

Bivalent vaccine group

Number at Risk	290292	265721	190506	92117	23394	0
Cumulative Events	0	298	574	797	855	855
Cumulative Incidence	0	0.1	0.23	0.38	0.47	0.47

≥ 2 monovalent mRNA vaccine group

Number at Risk	580584	377559	210906	62090	6448	0
Cumulative Events	0	1173	1851	2064	2083	2083
Cumulative Incidence	0	0.25	0.48	0.63	0.68	0.68

COVID-19 unvaccinated group

Number at Risk	204655	181313	126436	58604	14412	0
Cumulative Events	0	337	605	782	813	815
Cumulative Incidence	0	0.17	0.34	0.53	0.61	0.63

Figure 3. Cumulative incidence COVID-19 hospital death among the bivalent (original and Omicron BA.4/BA.5) mRNA-1273 COVID-19 vaccine cohort, ≥ 2 monovalent mRNA vaccine cohort, and COVID-19 unvaccinated cohort.

	Months of Follow-up					
	0	1	2	3	4	5
Bivalent vaccine group						
Number at Risk	290292	266013	190954	92500	23507	0
Cumulative Events	0	3	6	9	10	10
Cumulative Incidence	0	0	0	0	0.01	0.01
≥ 2 monovalent mRNA vaccine group						
Number at Risk	580584	378636	212132	62511	6494	0
Cumulative Events	0	24	51	59	59	59
Cumulative Incidence	0	0.01	0.01	0.02	0.02	0.02
COVID-19 unvaccinated group						
Number at Risk	204655	181634	126892	58935	14505	0
Cumulative Events	0	11	20	28	35	35
Cumulative Incidence	0	0.01	0.01	0.02	0.04	0.04

8) The authors may wish to consider including the STROBE checklist as a supplementary file. This addition would ensure that all essential items required for reporting have been appropriately included and adhered to in the study.

Response: Thanks for the comment. Per the Journal's guideline, we confirm that this observational study has been reported according to the STROBE statement. The STROBE checklist is provided in the Supplementary material.

References

- 1 Stowe, J., Andrews, N., Kirsebom, F., Ramsay, M. & Bernal, J. L. Effectiveness of COVID-19 vaccines against Omicron and Delta hospitalisation, a test negative case-control study. *Nat Commun* **13**, 5736 (2022).
- 2 Khoury, D. S. *et al.* Predicting the efficacy of variant-modified COVID-19 vaccine boosters. *Nat Med* **29**, 574-578 (2023).
- 3 Link-Gelles, R. *et al.* Early Estimates of Bivalent mRNA Booster Dose Vaccine Effectiveness in Preventing Symptomatic SARS-CoV-2 Infection Attributable to Omicron BA.5- and XBB/XBB.1.5-Related Sublineages Among Immunocompetent Adults - Increasing Community Access to Testing Program, United States, December 2022-January 2023. *MMWR Morb Mortal Wkly Rep* **72**, 119-124 (2023).
- 4 Surie, D. *et al.* Early Estimates of Bivalent mRNA Vaccine Effectiveness in Preventing COVID-19-Associated Hospitalization Among Immunocompetent Adults Aged ≥ 65 Years - IVY Network, 18 States, September 8-November 30, 2022. *MMWR Morb Mortal Wkly Rep* **71**, 1625-1630 (2022).
- 5 Barouch, D. H. Covid-19 Vaccines — Immunity, Variants, Boosters. *N Eng J Med* **387**, 1011-1020 (2022).
- 6 Keeton, R. *et al.* T cell responses to SARS-CoV-2 spike cross-recognize Omicron. *Nature* **603**, 488-492 (2022).

REVIEWERS' COMMENTS

Reviewer #1 (Remarks to the Author):

I would like to thank the authors for addressing my concerns and clarifying a number of points I raised. I have no other comments and recommend this manuscript of publication

Reviewer #2 (Remarks to the Author):

Thank you for these responses and edits. I have no further comments.